# Forecasting of NIFTY 50 Index Price by Using Backward Elimination with an LSTM Model

Syed Hasan Jafar [1], Shakeb Akhtar [1], Hani El-Chaarani [2,*], Parvez Alam Khan [3] and Ruaa Binsaddig [4]

[1] School of Business, Woxsen University, Hyderabad 502345, India; syedhasan.jafar@gmail.com (S.H.J.); shakebakhtar.amu@gmail.com (S.A.)

[2] Faculty of Business Administration, Beirut Arab University, Riad El Solh, Beirut 11072809, Lebanon

[3] Department of Management and Humanities, University Technology PETRONAS, Seri Iskandar 32610, Malaysia; parvezkhanalam@gmail.com

[4] College of Business Administration, University of Business and Technology, 10000 Prishtina, Kosovo; r.binsaddig@ubt.edu.sa

* Correspondence: h.shaarani@bau.edu.lb

**Abstract:** Predicting trends in the stock market is becoming complex and uncertain. In response, various artificial intelligence solutions have emerged. A significant solution for predicting the trends of a stock's volatile and chaotic nature is drawn from deep learning. The present study's objective is to compare and predict the closing price of the NIFTY 50 index through two significant deep learning methods—long short-term memory (LSTM) and backward elimination LSTM (BE-LSTM)—using 15 years' worth of per day data obtained from Bloomberg. This study has considered the variables of date, high, open, low, close volume, as well as the 14-period relative strength index (RSI), to predict the closing price. The results of the comparative study show that backward elimination LSTM performs better than the LSTM model for predicting the NIFTY 50 index price for the next 30 days, with an accuracy of 95%. In conclusion, the proposed model has significantly improved the prediction of the NIFTY 50 index price.

**Keywords:** backward elimination; LSTM; stock market prediction; NIFTY 50; relative strength index; accuracy score

## 1. Introduction

The recent advancements in smart tools can predict security prices using technical analysis and fundamental analysis (Maniatopoulos et al. 2023), and can also use derivatives data analysis, including open interest and put call ratio. The scope of the significant development of emerging technology in Fintech has acted as a beacon in finance (Weng et al. 2018; Gao et al. 2022). Investor confidence and investment quality are both enhanced by the tremendous research opportunities available in this area (Mondal et al. 2021). Research in this area is more often conducted by corporate entities, who use asset classes to forecast asset prices on back-tested data (Cui et al. 2023). While employing these techniques has helped predict future stock price, achieving maximum accuracy in the prediction is still a challenge. This is because the index or stock momentum depends on various factors like news flow, global and domestic market sentiment, geopolitical scenarios/tensions, FII and DII flow, domestic growth stimulating factors, regulatory body decisions and policy, central government and central bank policy, etc. However, the use of the NIFTY 50 price helps market participants make better judgments and improve strategies in the future and options (F&O) segment or in the cash market (Jain et al. 2018; Vineela and Madhav 2020). The NIFTY 50 is an Index of 50 listed companies that act as derivatives of underlying stock within the portfolio called the NIFTY 50 index (Mondal et al. 2021). The highly volatile and chaotic nature of the stock market creates variation and makes it unpredictable in terms of return generation, closing price, factors impact, and influence of price action factors

(Sheth and Shah 2023). The performance and return generated by the NIFTY 50 are directly proportioned to the performance/return of the underlying stock, considering that the weightage assigned to each underlying stock belongs to the NIFTY 50 index (Mondal et al. 2021). Monitoring the NIFTY 50 index enables traders and investors to manage the risk and reward ratio and point risk in the available market by calculating the ATR (average true range).

In recent years, there has been a growing interest in research employing artificial intelligence-based techniques for stock market prediction using the NIFTY 50 data, several machine learning models, including logistic regression (LR), support vector machine (SVM), random forest, etc., have been used for solving specific difficulties in time series forecasting (Abraham et al. 2022; Jin and Kwon 2021; Mehtab and Sen 2020; Parmar et al. 2018; Vijh et al. 2020). However, predicting the real-time market requires models to detect hidden data patterns in order to analyze such time-series data. While machine learning aids in discovering hidden patterns, it is not helpful for all-time series data (Idrees et al. 2019; Thakkar and Chaudhari 2021). The literature has also explored the neural networks method, but a simple neural network seems to be unable to predict market trends, and it even degrades the model's accuracy. A possible solution is the use of deep neural networks (Olorunnimbe and Viktor 2023), which examine data attributes and take historical data and fluctuations into account to solve this problem. Deep neural networks (DNN), convolutional neural networks (CNN), and long short-term memory networks (LSTM) are three deep neural models that have been efficiently used in the literature to predict stock prices (Ananthi and Vijayakumar 2021; Chen et al. 2021; Dash et al. 2019; El-Chaarani 2019). Among these aforementioned methods, LSTM has been employed in deep learning models for stock price prediction, and it has produced better results (Liu et al. 2021; Mehtab et al. 2020; Nelson et al. 2017; Polamuri et al. 2021; Rezaei et al. 2021; Shen and Shafiq 2020). Although these approaches are acknowledged to be highly useful in data investigation, accuracy in prediction becomes challenging when the time series data is highly unstable and stochastic.

The current study suggests a more accurate method to predict the NIFTY 50 price for the next 30 days by utilizing LSTM and LSTM with backward elimination. A comparison has been made between these two models to predict the closing price of the NIFTY 50 index, and the results are presented in this paper. To indicate the closing price, we have considered specific variables such as date, high, open, low, close volume, and 14-period relative strength index (RSI) values. The subsequent sections of the paper are organized as follows. Section 2 provides a concise overview of the existing research pertaining to the application of deep learning techniques in the prediction of the NIFTY 50. The proposed methodology is explained in Section 3. The discussion regarding the experimental data is presented in Section 4, while Section 5 provides the concluding remarks of the study, including an examination of its limitations and suggestions for future research.

## 2. Related Work

Predicting the stock price can be achieved using two methods. The first method is based on old models, such as the autoregressive integrated moving average (ARIMA) (Ilkka and Yli-Olli 1987) and the Cartesian autoregressive integrated moving average search algorithm (CARIMA) (Ostermark 1989). The second method is based on contemporary AI models, such as machine learning models (Parmar et al. 2018; Chen et al. 2021), artificial neural networks (Vijh et al. 2020), deep learning (Jiang 2021; Jing et al. 2021), fuzzy logic (Xie et al. 2021). Idrees et al. (2019), focusing on developing an effective ARIMA model for predicting the volatility of the Indian stock market based on time series data. Vaisla and Bhatt (2010) suggested the use of an analysis of the performance of the artificial neural network technique for stock market forecasting. The projected time series was compared to the actual time series, which showed a mean percentage error of about 5% for both the NIFTY 50 and the Sensex, on average. Validation of the anticipated time series may be performed using a variety of tests. However, for the sake of validation, we employed the

ADF and the Ljung–Box tests in this work. We believe that the ARIMA method is adequate for dealing with time-series data, but the drawbacks of choosing the variables were not studied, and the accuracy rate was not calculated for that model.

The NIFTY 50 is an index of 50 listed companies that act as a derivative of underlying stock within the portfolio called the NIFTY 50 index (Mondal et al. 2021). Kurani et al. (2023) have used an artificial neural network to forecast stock values in the financial industry. The authors also investigated the influence of various microeconomics variables and physical elements on the stock price of different financial sector stock values in the financial industry (Kurani et al. 2023). The proposed ANN model yields a maximum error rate of 16.13% for an Axis Bank stock. However, when the macroeconomics factors are boosted, it results in a decrease in the error rate (Jain et al. 2018).

Implementing AI models in predicting the stock prices gradually increases the model's learning ability. In their work, Dash et al. (2019) have made a comparisons between individual classifiers and various ensemble models. A total of 13 classifiers are ranked using the TOPSIS approach, including 7 original classifiers, i.e., radial basis function network, k-nearest neighbor (KNN), support vector machine (SVM), decision tree (DT), logistic regression (LR), naive Bayes (NB), and multilayer perceptron (MLP), as well as and 6 alternate models, i.e., accuracy (A), precision (P), recall (R), f-measures (F1), true positive, true negative, and G-mean. According to the findings, the TOPSIS-based base classifier is used for the CS ensemble to yield more accurate predictions than other ensemble models. This technique also aids in picking the best-approaching classifiers for this model. Long et al. (2019) suggested a multi-filters neural network for the feature engineering of multivariate financial time series and classification-based prediction using a deep learning approach. Compared to RNN, CNN, and other machine learning models, the prediction result from the MFNN surpassed those of the best machine learning technique, with an accuracy of 55.5%, which was the most accurate prediction. Long et al. (2019) have advised using a particular network to harvest data from many sources (macroeconomic indicators, news, and market emotion) for better predictions. Vijh et al. (2020) have employed artificial neural network and random forest approaches to predict the closing price of five distinct company sectors. They predicted stock closing prices using the RMSE, MBA, and MAPE indicators. The estimated RMSE, MAPE, and MBE indicators in this research show that ANN outperforms RF in forecasting stock prices. In their study, Ananthi and Vijayakumar (2021) used the k-NN regression method to forecast market trends. The stock prices of numerous firms are evaluated, and a collection of technical indicators are projected. The results revealed a significant increase in accuracy between 75% and 95% compared to other machine learning techniques.

Chen et al. (2021) have combined XGboost with an enhanced firefly algorithm for stock price prediction and a mean-variance model for portfolio selection to create a hybrid model. The suggested model was tested on the Shanghai Stock Exchange and was found to be highly efficient in terms of returns and risks. Selvamuthu et al. (2019) utilized neural networks (NN) based on three distinct approaches, namely the Levenberg–Marquardt (LM) approach, scaled conjugate gradient (SCG), and Bayesian regularization (BR), to forecast Indian stock market movements using tick data and15-minute data from an Indian firm, comparing the outcomes. In this case, all these algorithms achieved an accuracy rate (A) of 99.9% when using tick data. The accuracy rate for all these models across a 15-minute dataset drops to LM, 96.2%; SCG, 97.0%; and BR, 98.9%; respectively, which was much lower than the results achieved using tick data. In another study, Mehtab et al. (2020) used eight machine-learning models and four deep learning approaches to provide numerous ways to predict stock-index values and price-movement patterns on a weekly-forecast prospect. The predicted models were built, optimized, and tested using the NIFTY 50 index values from 29 December 2014 to 31 July 2020. The performance of the LSTM-based regression models was shown to be considerably better than that of the machine learning-based prediction models. In another study, in 2020, Vineela and Madhav carried out a study regarding the closing price of chosen stocks, including HDFC, HDFC Bank, Reliance,

TCS, Infosys, Bharti Airtel, HUL, ITC, Kotak Mahindra, and ICICI Bank, in which they projected the stock price for 60 days using the LSTM model. The influence of the NIFTY 50 index on selected stocks was also explored. Except for the dependence stock, the other chosen stocks were expected to have an upward or stable trend over the next 60 days. It was also discovered that all the stocks chosen had a favorable correlation with the NIFTY 50 index. In a similar study, Parmar et al. (2018) attempted to predict future stock values by comparing regression and LSTM-based machine learning for predicting stock prices using fewer variables, such as open, close, low, high, and volume. Finally, it was discovered that LSTM is more efficient than regression models, with 87.5% accuracy vs. 86.6% accuracy.

A study on LSTM using new data was carried out by Sarode et al. (2019). In the study, a decision-making algorithm was built, based on historical data and news. The authors suggest that incorporating current tactics into existing quant trading strategies will motivate quant traders to invest and optimize their profits. The disadvantage in this study is that it is merely research, not an experimental investigation, and there is no current data. It also indicates that multivariate investigation is not a good method in regards to LSTM models, since univariate approaches are more accurate and quicker to run. Again, Mehtab and Sen (2020) have used eight regression and eight classification algorithms to demonstrate numerous stock-value and movement (up/down) prediction approaches on a weekly-forecast prospect. These models are based on deep learning (DL) and machine learning (ML) techniques. These models were constructed, fine-tuned, and then evaluated using daily historical data from the NIFTY 50 from 5 January 2015 to 27 December 2019. The prediction context is further enhanced by building three CNN models, using univariate and multivariate techniques with varied input data sizes and network formations. This CNN-based approach outperformed machine-learning-based prediction models by a wide margin. The disadvantage is that there are many fluctuations in real-time stock prices. However, a share market combines a deep learning method with sentiment analysis. The main contribution of this method is the merging of the long short-term memory (LSTM) neural network technique for stock prediction with the convolutional neural network model for sentiment analysis. When the suggested model is compared to existing deep neural networks, it is observed that the proposed system has a low average MAPE of 0.0449. Long short-term memory and convolution neural networks are the models that produced the strongest outcomes in the stock market investigation. Rezaei et al. (2021) proposed two models: empirical mode decomposition (EMD) and complete ensemble empirical mode, combined with CNN and LSTM, in this research. The root mean square error, mean absolute error, and mean absolute percentage error assessment metrics were employed in this work. According to the experimental findings, combining CNN with LSTM gives better results than those obtained using other approaches. However, this approach does not incorporate the process of selection.

Due to its inherent complexity and uncertainty, stock market forecasting frequently draws criticism. Some claim that focusing entirely on technical analysis and historical data might ignore other important external influences. Predictions are unreliable because stock prices, according to the efficient market hypothesis, already reflect all available information. Additionally, it is possible for human nature and market emotion to be unexpected and unreasonable. The possibility for bias and overfitting in prediction algorithms is highlighted by critics. Although improvements in machine learning have increased its accuracy, doubts continue to exist over its capacity to consistently anticipate market moves, given the speculative nature of stock forecasting.

Based on the literature review conducted, it is evident that previous research in this domain has used various methodologies to forecast stock prices. However, there has not yet been a study on the approach of identifying the more correlated features before passing them into the LSTM model for stock price prediction. To identify the best-corelated features, the backward elimination with LSTM (BE-LSTM) method for predicting the stock price was built and compared with the general LSTM, without the backward elimination approach, in order to identify which model is superior.



Table 1 represents a more detailed review of the various machine learning models, particularly the LSTM model for predicting stock prices.

**Table 1.** Various methodologies for predicting stock price—a review.

| Year | Research Work | Methodology | Findings | Accuracy |
|------|---------------|-------------|----------|----------|
| 2017 | (Nelson et al. 2017) | LSTM | This suggested model has a lower risk than other models, when it comes to predicting the stock price. | 59.5% |
| 2018 | (Zhang et al. 2018) | unsupervised heuristic algorithm | This model will perform better in the future by considering the feature selection methods. | |
| 2018 | (Parmar et al. 2018) | regression model and LSTM model | The LSMT model is superior when compared to the regression model. | Regression: 86.6% LSTM: 87.5% |
| 2018 | (Jain et al. 2018) | artificial neural network | The error rate is high; more macroeconomic variables are required to reduce the error rate. | |
| 2019 | (Dash et al. 2019) | TOPSIS crow search-based weighted voting classifier ensemble | The ensemble methods perform well, but the predicted values are not close to the original values. | 84.3% |
| 2019 | (Idrees et al. 2019) | ARIMA model | The ARIMA method is adequate for dealing with time-series data. The drawback is that choosing the attributes are not chosen, and accuracy is not calculated for that model. | Ljung–Box test results (NIFTY) $p$-value = 0.9099 Ljung–Box test results (Sensex) $p$-value = 0.8682 |
| 2019 | (Long et al. 2019) | multi-filters neural network (MFNN) | Compared with RNN, CNN, LSTM, SVM, LR, RF, and LR, this proposed MFNN model performs well. The drawback is that it has minimal accuracy. | 55.5% |
| 2019 | (Sarode et al. 2019) | LSTM | Identifying which stock to invest in by analyzing historical data along with world news. The drawback is that it is only a study, not an experimental analysis, and lacks news data. | |
| 2019 | (Selvamuthu et al. 2019) | neural networks based on three different learning algorithms, i.e., Levenberg–Marquardt, scaled conjugate gradient, and Bayesian regularization | The error is high compared with the original stock price value. | 96.2%—LM, 97.0%—SCG, and 98.9%—Bayesian regularization |
| 2020 | (Mehtab and Sen 2020) | boosting, decision tree, random forest, bagging, multivariate regression, SVM and MARS algorithms | The multivariate regression algorithm is best when compared to other algorithms; the drawback is that it cannot be used with LSTM regression; it is not a generic model | 99% |
| 2020 | (Mehtab et al. 2020) | classification algorithm, KNN, boosting, decision tree, random forest, bagging, multivariate regression, SVM, ANN, and CNN algorithms | In this method, CNN with multivariate regression is better than CNN with univariate regression and other machine learning algorithms. Feature selection is not carried out in this study; hence, the possibility of biase is high. It is not a generic model. | 97% |
| 2020 | (Shen and Shafiq 2020) | feature engineering RE and RFE with LSTM for Chinese stock market data; less historical data | The ccuracy varies based on different PCA values. | 96% |

**Table 1.** *Cont.*

| Year | Research Work | Methodology | Findings | Accuracy |
|------|---------------|-------------|----------|----------|
| 2020 | (Vijh et al. 2020) | ANN and random forest | ANN is best when compared to a random forest classifier. | **Nike**<br>**ANN:**<br>RMSE—1.10<br>MAPE—1.07%<br>MBE——0.0522<br>**RF:**<br>RMSE—1.10<br>MAPE—1.07%<br>MBE——0.0522<br>**Goldman Sachs**<br>**ANN:**<br>RMSE—3.30<br>MAPE—1.09%<br>MBE—0.0762<br>**RF:**<br>RMSE—3.40<br>MAPE—1.01%<br>MBE—0.0761<br>**J.P. Morgan and Co.**<br>**ANN:**<br>RMSE—1.28<br>MAPE—0.89%<br>MBE——0.0310<br>**RF:**<br>RMSE—1.41<br>MAPE—0.93%<br>MBE——0.0138<br>**Pfizer Inc.**<br>**ANN:**<br>RMSE—0.42<br>MAPE—0.77%<br>MBE——0.0156<br>**RF:**<br>RMSE—0.43<br>MAPE—0.8%<br>MBE——0.0155 |
| 2020 | (Vineela and Madhav 2020) | LSTM | The stock prices of HDFC, HDFC Bank, Reliance, TCS, Infosys, Bharti Airtel, HUL, ITC, Kotak Mahindra, and ICICI Bank were forecasted for the next 60 days using the LSTM model.<br>It was also discovered that all of the stocks chosen had a favorable correlation with the NIFTY 50 Index. | Correlation percentage of selected stocks with NIFTY 50<br>HDFC—93%,<br>HDFC Bank—94%,<br>Reliance—86%,<br>TCS—94%,<br>Infosys—90%,<br>Bharti Airtel—51%,<br>HUL—92%,<br>ITC—79%,<br>Kotak Mahindra—97%,<br>and ICICI Bank—90% |
| 2021 | (Ananthi and Vijayakumar 2021) | KNN and candlestick regression | The price of the selected stocks was predicted using different machine learning algorithms, such as k-NN regression, linear regression, and support vector machine.<br>KNN performs well when compared with other algorithms. | Accuracy varies from 75% to 95%, based on the training dataset. |
| 2021 | (Chen et al. 2021) | XGBoost with IFA and mean-variance model | XGBoost with IFA was used for stock price prediction, with the mean-variance method employed for portfolio selection. | |
| 2021 | (Jing et al. 2021) | CNN-LSTM | CNN-LSTM performs, well with low average MAPE, compared to other deep neural networks. | Average MAPE of CNN-LSTM is 0.0449. |

**Table 1.** *Cont.*

| Year | Research Work | Methodology | Findings | Accuracy |
|---|---|---|---|---|
| 2021 | (Jin and Kwon 2021) | chart image | Compared to other methods, such as CNN, LSTM, PCA, MLP, the proposed method is superior. | 64.3% |
| 2021 | (Liu et al. 2021) | LSTM + social media news | A social media news attribute combined with an LSTM model is used for predicting the stock price. | 83% |
| 2021 | (Polamuri et al. 2021) | generative adversarial networks | The GAN-HPA algorithm beats the current MM-HPA model. MMGAN-HPA, on the other hand, improved the GAN-HPA. | 82% |
| 2021 | (Rezaei et al. 2021) | EMD-CNN-LSTM and EMD-LSTM | Applied for S&P 500, Dow Jones, DAX, and Nikkei225. | **S&P 500** **EMD-CNN-LSTM** RMSE—14.88 MAE—12.04 MAPE—0.611 **EMD-LSTM** RMSE—15.51 MAE—12.60 MAPE—0.639 **DOW JONES** **EMD-CNN-LSTM** RMSE—163.56 MAE—120.97 MAPE—0.6729 **EMD-LSTM** RMSE—171.40 MAE—128.55 MAPE—0.7184 **DAX** **EMD-CNN-LSTM** RMSE—108.56 MAE—86.05 MAPE—0.907 **EMD-LSTM** RMSE—109.97 MAE—86.75 MAPE—0.920 **Nikkei225** **EMD-CNN-LSTM** RMSE—194.17 MAE—147.18 MAPE—0.9413 **EMD-LSTM** RMSE—213.45 MAE—164.08 MAPE—1.0513 |
| 2021 | (Ribeiro et al. 2021) | HAR-PSO-ESN model | HAR-PSO-ESN is the model that was built. It is compared to current requirements, such as the autoregressive integrated moving average, HAR, multilayer perceptron (MLP), an ESN, with predicting possibilities of 1 day, 5 days, and 21 days. The predictions are compared using r-squared and mean-squared error performance metrics, followed by a Friedman test and a post-hoc Nemenyi test. | Average $R^2$ (coefficient of 1 day—0.635, 5 days—0.510, and 21 days—0.298, and average mean squared error of 1 day—5.78 10 8, 5 days—5.78 10 8, and 21 days—1.16 10 7. |

**Table 1.** *Cont.*

| Year | Research Work | Methodology | Findings | Accuracy |
|------|---------------|-------------|----------|----------|
| 2021 | (Xie et al. 2021) | Hammerstein–Wiener model | The nonlinear input and output nonlinearities of the Hammerstein–Wiener model are substituted with the fuzzy system's nonlinear fuzzification and defuzzification processes, allowing the inference processes to be interpreted using fuzzy linguistic rules derived from linear dynamic computing. Three financial stock datasets are used to test the efficacy of the proposed model. | **S&P 500** MAE—$1.39 \times 10^2$ RMSE—$1.79 \times 10^2$ NRMSE—0.242 **HSI** MAE—$3.99 \times 10^3$ RMSE—$4.76 \times 10^3$ NRMSE—0.808 **DJI** MAE—$8.71 \times 10^3$ RMSE—$9.78 \times 10^3$ NRMSE—1.931 |
| 2022 | (Sisodia et al. 2022) | deep learning LSTM | The NIFTY 50 stock price statistics over 10 years are used. The data was collected from 2011 to 2021. Normalized data is utilized for model training and testing. | A promising 83.88% accuracy for the proposed model. |
| 2022 | (Mahajan et al. 2022) | LSTM models with GARCH and RNN | In NIFTY 50 volatility prediction, GARCH- and RNN-based overall GARCH models are marginally better than RNN-based LSTM models. | Both models have similar accuracy. |
| 2023 | (Zaheer et al. 2023) | CNN, RNN, LSTM, CNN-RNN, and CNN-LSTM. | With the exception of CNN, the model outperformed all other models. | CNN-LSTM-RNN has the highest accuracy of 98%. |
| 2023 | (Sharma et al. 2023) | Five stock price prediction algorithms that are used: random forest, SVR, ridge, lasso regression, and the KNN model. | Support vector regression (SVR) performs more accurately than the lasso and random forest, KNN, and the ridge model. | Support vector regression 83.88% |
| 2023 | (Oukhouya and El Himdi 2023) | SVR, XGBoost, MLP, and LSTM | The support vector regression (SVR) and multilayer perceptron (MLP) models exhibit superior performance compared to the other models, showing high levels of accuracy in predicting daily price fluctuations. | SVR Accuracy 98.9% |
| 2023 | (Mahboob et al. 2023) | MLS LSTM | This study develops a unique optimization method for forecasting stock prices, employing an MLS LSTM model and the Adam optimiser. | MLS LSTM accuracy 95.9% |
| 2023 | (Bathla et al. 2023) | LSTM | Using mean absolute percentage error (MAPE) values demonstrates greater accuracy than using conventional data analytics methodologies. | LSTM accuracy 90% |

In addition, the recent literature LSTM stock prediction approach has also been used, with an accuracy range of 83–90% (Bathla et al. 2023; Mahajan et al. 2022; Sisodia et al. 2022), and the support vector regression (SVR) and multilayer perceptron (MLP) models show superior performance compared to the other models, exhibiting levels 98.9% accuracy in predicting daily price fluctuations. The SVR (support vector regression) accuracy of 98.9% outperforms the lasso and random forest method, followed by the KNN and ridge model (Oukhouya and El Himdi 2023). Furthermore, the MLS LSTM, and CNN-LSTM-RNN have the highest accuracy range from 95–98.9% (Mahboob et al. 2023; Zaheer et al. 2023). The detailed literature reviews indicate that the LSTM method is not able to predict the stock price with a high rate of accuracy. Therefore, long short-term memory (LSTM) and backward elimination LSTM (BE-LSTM) seems promising for high accuracy in forecasting

the stock price. The following section discusses the methods and approaches adopted to achieve the research objective.

## 3. Methodology

The proposed work is a new learning-based approach for NIFTY 50 price forecasting. Backward elimination using LSTM (BE-LSTM) is the primary mechanism used in this study. Meanwhile, to understand the suggested method, it is crucial to first comprehend what backward elimination (BE) and LSTM are and how they will perform. Hence, a brief description of these methods is provided in the subsequent section.

### 3.1. Data Collection

This study is inclined to predict the closing price of the NIFTY 50 index, considering historical data. The data range selected for the study is taken from the Bloomberg database, starting from 11 February 2005 and ending on 5 March 2021. The data consideration includes approximately 15 years of data, which consist of bull and bear phases of the Indian equity market for better analysis and prediction. The study required a historical dataset from a reliable source and an input data which should be relevant and appropriate for the upcoming price prediction. The present study used Bloomberg, the most trusted data source in finance, which provides historical security data in the required form. The study used 15 years of technical analysis data, including HOLC, i.e., high, open, low, and close of daily trade. The data points considered are NIFTY 50 daily volume and 14 periods of RSI as an indicator. The above data was used to train the model to predict the closing price of the NIFTY 50.

### 3.2. Data Pre-Processing

The data pre-processing is essential in determining the data fit to the trained model in order to obtain the NIFTY 50 price prediction. The process involves removing duplicate data and avoiding the related missing data. The NIFTY 50 dataset of 15 years was split into 80% for training the model and 20% for testing. The model is then set to segregate the data into training and validation data types. This is a feature selection technique used to build a predictive model. The primary use of this algorithm is to eliminate features that do not have any correlation with the dependent variable or prediction of the output. The process of backward elimination is explained in Table 2.

**Table 2.** Algorithm—backward elimination process.

| | |
|---|---|
| Step-1 | Initially, we need to see obtain significance level (SI = 0.05) in the model. |
| Step-2 | Fit the model with all independent variables. |
| Step-3 | Choose the independent variable which has the highest *p*-value. If *p*-value > significance level (SL), then it continues to step 4. Otherwise, it terminates. |
| Step-4 | Remove that independent variable. |
| Step-5 | Rebuild and fit the model with the remaining featured variable. |

The probability value is defined as the *p*-value. It is used as a substitute for the point of rejection to manifest the low significant value, in which the null hypothesis would reject it. If the value of *p* is less than 0.05, then the evidence for the alternative view will become stronger.

$$\text{Teststatic (Z)} = \frac{S - S_0}{\sqrt{\frac{S_0(1 - S_0)}{n}}} \tag{1}$$

where $S$ is the sample proportion, $S_0$ is the proportion of the assumed population in the null hypothesis, and $n$ is the sample size. The *p*-value level can be obtained from the obtained Z value.

Step-by-step breakdown of the process.

Step 1. Formulate the Hypotheses:

- Null hypothesis (H0): The proportions are equal; $S = S_0$.
- Alternative hypothesis (H1): The proportions are not equal; $S \# S_0$.

Step 2. Calculate the Sample Proportions:

Calculate the sample proportions $S$ and $S_0$:

- $S$ is the proportion of success in the sample.
- $S_0$ is the hypothesized proportion of success (given in the null hypothesis).

Step 3. Calculate the Standard Error:

The standard error (SE) of the difference between two proportions can be calculated as:

$$\text{SE} = \sqrt{\frac{S(1 - S)}{n} + \frac{S_0(1 - S_0)}{n_0}}$$

where $n$ is the sample size, and $n_0$ is the reference sample size.

Step 4. Calculate the Z-score:

The Z-score measures the difference between the number of standard errors observed between the sample proportions and the expected difference under the null hypothesis. It is calculated as: $Z = S - S_0/\text{SE}$.

Step 5. Determine the Critical Value or *p*-value:

Depending on the selected significance level ($\alpha$), determine the critical value by referencing the standard normal distribution table. Alternatively, you can compute the *p*-value linked to the Z-score using the standard normal distribution.

Step 6. Make a Decision:

If using critical values, compare the calculated Z-score to the critical value. If using *p*-values, compare the *S*-value to your chosen significance level ($\alpha$). If the *S*-value is less than $\alpha$, reject the null hypothesis. If the *S*-value is greater than or equal to $\alpha$, accept the null hypothesis.

Step 7. Interpretation:

If the null hypothesis is rejected, this suggests that there is a significant difference between the proportions $s$ and $s_0$. If the null hypothesis is not rejected, it means that there is not enough evidence to conclude that the proportions are significantly different.

*3.3. LSTM Model*

Hochreiter has designed long short-term memory (LSTM) to overcome speed and stability problems in recurrent neural networks (RNN). It can retrieve data from the beginning of time and utilize it to make future predictions. The vector length assigned to the node is 64, and there is just one hidden layer in a neural network. The data dimensions determine the number of nodes in the input layer. The input layer's nodes may be linked to the concealed layer's nodes through synapses. The weight is a coefficient in the relationship between the input and the concealed node—a signal decision maker (Ribeiro et al. 2021; Selvamuthu et al. 2019). The modification of weights is a normal part of the learning process. The artificial neural network will assign ideal weights for each synapse when the learning process is completed. The nodes of the hidden layer, with activation functions such as sigmoid, ReLU, or the tangent hyperbolic (tanh) function, will determine whether that node should be activated or not. This conversion will provide data with the lowest error value when comparing the trained model and test model, if the softmax function is used. The NN output layer comprises the values received after the transformation (Xie et al. 2021). If the results obtained are not optimal, the back propagation procedure can be

used. The back propagation (BP) technique will update the weights of the hidden layers, sending the information from the output that reduces the error across the given set of epochs (Nelson et al. 2017; Mehtab et al. 2020; Liu et al. 2021).

This approach may be repeated to improve forecasts and minimize prediction errors. The model obtained will be trained after this procedure is completed. Recurrent neural networks are neural networks that anticipate future values based on previous observation sequences (RNN). This type of NN makes use of previously learned data to estimate future trends. These stages of previous data should be memorized to anticipate and guess future values. In this case, the hidden layer serves as a repository for primary data from the sequentially acquired data. The term "recurrent" can be used to describe the process of forecasting future data using previous portions of sequential data.

RNN cannot store memory for long (Shen and Shafiq 2020). The usage of long short-term memory (LSTM) proved to be very useful in foreseeing cases with long-time data based on "memory line". In LSTM, the earlier memorization stage can be performed through gates by incorporating memory lines. Each node is substituted with LSTM cells in hidden layers. Each cell is equipped with a forget gate ($e_t$), an input gate ($j_t$), and an output gate ($m_t$). The functions of the gates are as follows: the forget gate is used to eradicate the data from the cell state, the input gate is used to add data to the cell state, and the output gate holds the output of the LSTM cell, as shown in Figure 1.

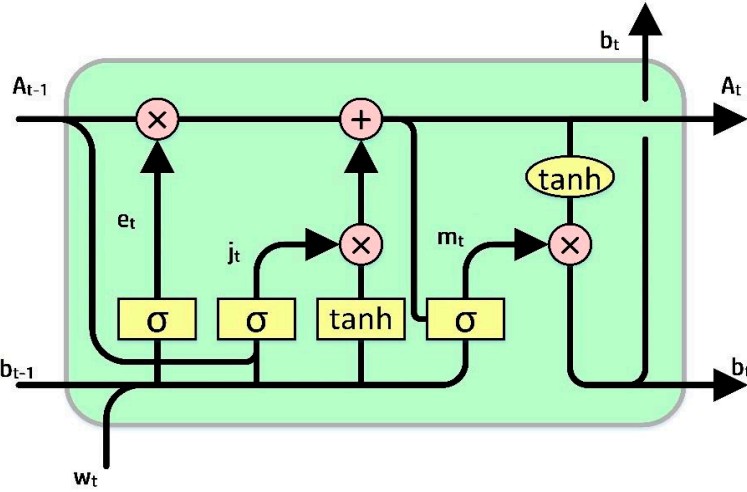

**Figure 1.** A sample representation of the LSTM model (Sezer et al. 2020).

The goal is to control the state of each cell. The forget gate ($e_t$) can output a number between 0 and 1. When the output is 1, it signals to hold the data, whereas a 0 signals to ignore the data, and $e_t$ represents the vector values ranging from 0 to 1, corresponding to each number in the cell, $A_{t-1}$.

$$e_t = \sigma(P_e[b_{t-1}, W_e] + q_e) \tag{2}$$

In Equation (2) Pe represents the weight matrix associated with the forget gate, and $\sigma$ is the sigmoidal function. The memory gate ($j_t$) chooses the data to be stored in the cell. The sigmoid input layer determines the values to be changed. After that, a tanh layer adds a new candidate to the state. The output gate ($m_t$) determines the output of each cell. The output value will be based on the state of the cell, along with the filtered and freshest data.

$$j_t = \sigma(P_j[b_{t-1}, W_j] + q_j) \tag{3}$$

$$m_t = \sigma(P_m[b_{t-1}, W_m] + q_m) \tag{4}$$

$$b_t = m_t \tanh(A_t) \tag{5}$$

where $W_e$, $W_j$, and $W_m$ are weight matrices, $q_e$, $q_j$, and $q_m$ are bias vectors, $b_t$ is the memory cell value at time t, and $e_t$ corresponds to the forget gate value. Whereas, Pj represents the weight matrix associated with the input gate, and Pm represents the weight matrix associated with the output gate. $A_t$ represents the current cell state, the input gate value is represented by $j_t$, and $m_t$ represents the output gate value.

### 3.4. Backward Elimination with LSTM (BE-LSTM)

LSTMs are incredibly effective in solving sequence prediction problems because they can retain old data. Hence, LSTM can be a good choice for our prediction problem, as the historical price is vital in determining its future price. In related research, the LSTM model was employed for predicting the stock price. Figure 2 represents the processing stages in developing an LSTM-based stock prediction model, and the algorithm for designing the LSTM model is given in Table 3.

**Table 3.** Algorithm—building the LSTM model.

| | |
|---|---|
| Step-1 | Import the libraries, such as Pandas, Tensor Flow, Sequential, LSTM, Dense, Dropout, and Adam |
| Step-2 | Import the data and pre-processing the data for identifying and handling the missing values, encoding the categorical data, splitting the dataset, and feature scaling. |
| Step-3 | Create an LSTM model with input, hidden, and output layers. |
| Step-4 | Compile the LSTM model and fitting the data. |
| Step-5 | Calculate the error and accuracy of the model. |

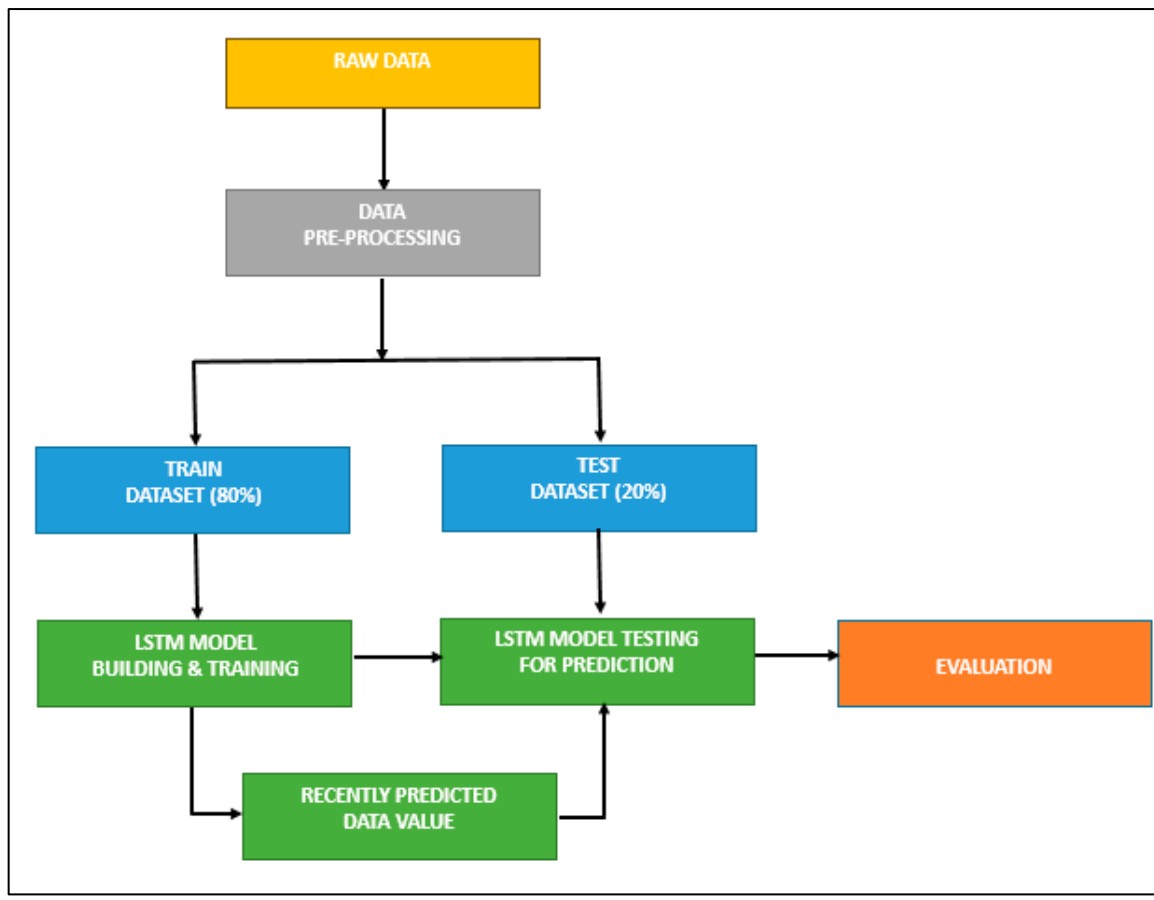

**Figure 2.** LSTM model without feature selection.

In the proposed method, the backward elimination method has been used as a feature selection method, and it is performed after the data pre-processing stage (Figure 3). This is done to determine which independent variable has a high correlation with the dependent

variable (date, open, high, low, close, volume, value, trades, RSI, and RSI average). The selected variables are taken as inputs and sliced into training and test sets. Finally, they are entered into the LSTM model for prediction. A brief description of the BE-LSTM algorithm is given in Table 4. The backward elimination method is expected to decrease computational complexity and increase accuracy.

**Table 4.** Algorithm—building the BE-LSTM model.

| | |
|---|---|
| Step-1 | Import the libraries, such as Pandas, Tensor Flow, sequential, LSTM, Dense, Dropout, and Adam. |
| Step-2 | Import the data and pre-processing the data for identifying and handling the missing value, encoding the categorical data, splitting the database, and feature scaling. |
| Step-3 | Initially, we need to set the significance level (SL = 0.05) in the model. |
| Step-4 | Fit the model with all independent variables. |
| Step-5 | Choose the independent variable which has the highest *p*-value. If *p*-value > significancelLevel (SL), then it progresses to Step-4. Otherwise, it terminates. |
| Step-6 | Remove that independent variable and repeat Step-5 until the *p*-value is not greater than 0.05. |
| Step-7 | Split the training and test data between the remaining featured variables. |
| Step-8 | Create an LSTM model with input, hidden, and output layers. |
| Step-9 | Compile the LSTM model and fit the data. |
| Step-10 | Calculate the error and accuracy of the model. |

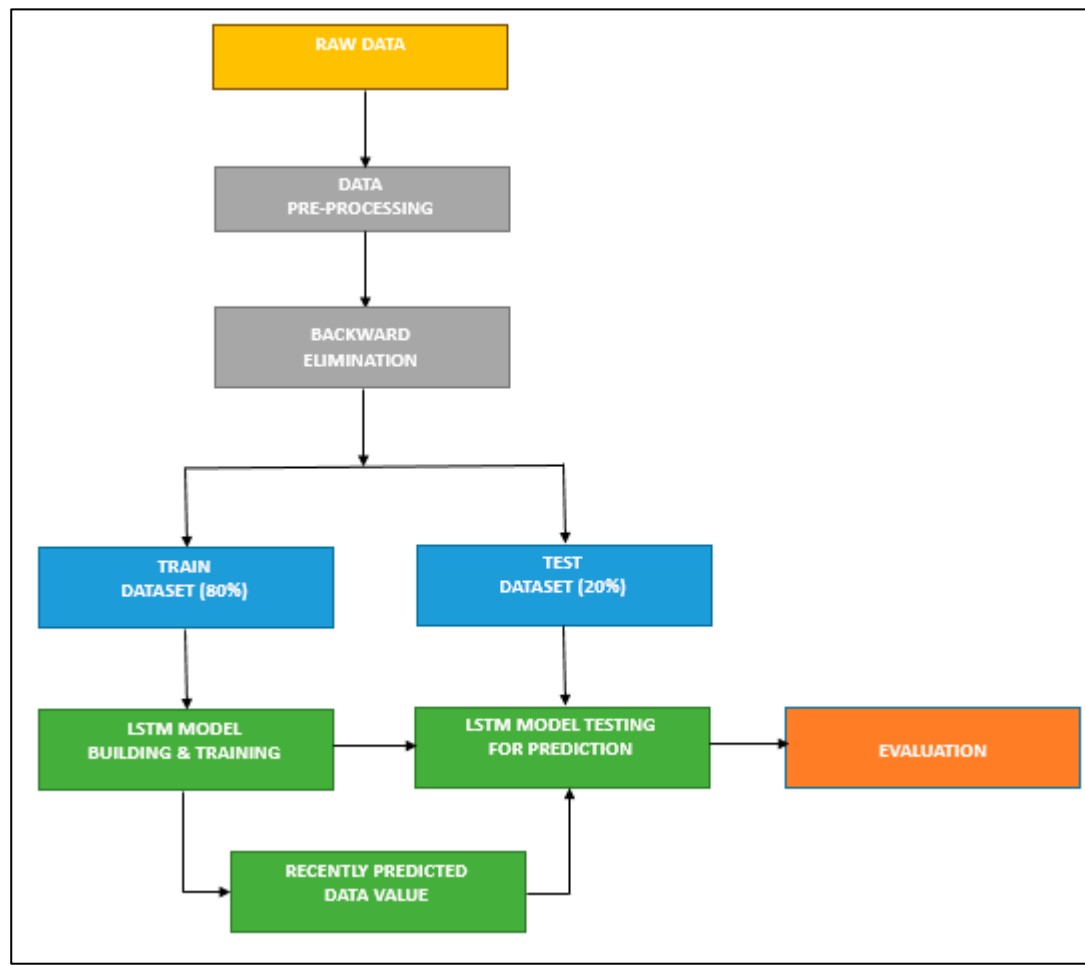

**Figure 3.** BE—LSTM model.

*3.5. Evaluation Metrics*

Mean square error (MSE), root mean square error (RMSE), and mean absolute percentage error (MAPE) are used to assess the performance of the proposed LSTM- and

BE-LSTM-based model (Rezaei et al. 2021; Polamuri et al. 2021). The following is the formula for these metrics:

$$\text{MSE} = \frac{1}{m} \sum_i^m \left( b_i - \dot{b}_i \right)^2.$$ (6)

$$\text{RMSE} = \sqrt{\frac{1}{m} \sum_i^m \left( b_i - \dot{b}_i \right)^2}.$$ (7)

$$\text{MAPE} = \frac{1}{m} \sum_{i=1}^m \frac{\left| b_i - \dot{b}_i \right|}{b_i} \times 100.$$ (8)

**Accuracy**

Accuracy serves as a metric that offers a broad overview of a model's performance across all classes. It proves particularly valuable when all classes have equal significance. This metric is computed by determining the proportion of correct predictions in relation to the total number of predictions made.

$$\text{Accuracy} = \frac{\text{True}_{\text{positive}} + \text{True}_{\text{negative}}}{\text{True}_{\text{positive}} + \text{True}_{\text{negative}} + \text{False}_{\text{positive}} + \text{False}_{\text{negative}}}$$

Calculating accuracy using scikit-learn, based on the previously computed confusion matrix, is performed as follows: we store the result in the variable 'acc' by dividing the sum of true positives and true negatives by the sum of all the values within the matrix.

**Precision**

Precision is computed by taking the proportion of correctly classified positive samples relative to the total number of samples classified as positive, whether correctly or incorrectly. Precision serves as a metric for gauging the model's accuracy when it comes to classifying a sample as positive.

$$\text{Precision} = \frac{\text{True}_{\text{positive}}}{\text{True}_{\text{positive}} + \text{False}_{\text{positive}}}$$

When the model generates numerous incorrect positive classifications, or only a few correct positive classifications, this elevates the denominator and results in a lower precision score. Conversely, precision is higher under the following conditions:

1. The model produces a substantial number of correct positive classifications, thus maximising the true positives.
2. The model minimizes the number of incorrect positive classifications, thereby reducing false positives.

**Recall**

Recall is determined by the ratio of correctly classified positive samples to the total number of positive samples. It quantifies the model's capacity to identify positive samples. A higher recall score signifies a greater ability to detect positive samples.

$$\text{Recall} = \frac{\text{True}_{\text{positive}}}{\text{True}_{\text{positive}}, + \text{False}_{\text{negative}}}$$

Recall exclusively focuses on the classification of positive samples and is independent of the classification of negative samples, as observed in precision. If the model categorizes all positive samples as positive, even if it incorrectly labels all negative samples as positive, the recall will still register at 100%.

## 4. Results and Discussion

The historical data of NIFTY 50 was extracted from Yahoo Finance. The period of data covers from 20 January 2005 to 5 March 2021. It consists of 3986 data points and 8 attributes. The attributes are date, open, high, low, volume, value, trades, and RSI average (detail is shown in Table 5). By utilizing backward elimination (BE), this study has identified which independent variable is significantly correlated with the dependent variable.

**Table 5.** Detail of attributes.

| Constant | Attributes/Column Name |
|---|---|
| X1/Beta1 | DATE |
| X2/Beta2 | OPEN |
| X3/Beta3 | HIGH |
| X4/Beta4 | LOW |
| X5/Beta5 | VOLUME |
| X6/Beta6 | VALUE |
| X7/Beta7 | TRADES |
| X8/Beta8 | RSI |

In the case of backward elimination, we are now attempting to remove less important variables from the model. It usually entails repeatedly fitting the model, determining each variable's importance, and eliminating the least relevant variables.

In essence, we enable the model to include an intercept term constant that reflects the projected value of y when all independent variables are set to zero by including a constant term (a column of 1s) in the dataset. When the independent variables have no influence, the baseline level of y is captured by this intercept term.

When a variable is removed from the model, we are effectively determining its relevance by observing how the overall model performance (often evaluated by a metric like *p*-value, AIC, or R-squared) changes; therefore, adding this constant term is very important during backward elimination. Without the constant term, eliminating a variable can lead to a model that assumes the dependent variable starts at zero in the absence of all other factors, which may not be applicable in many real-world cases.

Initially, we need to confirm all the independent variables in the backward elimination algorithm, as shown in Table 6.

**Table 6.** Backward elimination (Step 1).

| DEP.Variable | | Close | | R-Squared: | | 1.000 |
|---|---|---|---|---|---|---|
| Model: | | OLS | | ADJ. R-Squared: | | 1.000 |
| Method | | Least Squares | | F-statistic: | | 4.785 |
| Date: | | Sat, 31 July 2021 | | Prob (F-Statistic): | | 0.00 |
| Time: | | 12:20:45 | | Log-Likelihood: | | $-19,235$ |
| No. observations: | | 3986 | | AIC: | | $3.849 \times 10^4$ |
| Df Residuals: | | 3977 | | BIC: | | $3.854 \times 10^4$ |
| Df Model | | 8 | | | | |
| Covariance Type: | | Non-robust | | | | |
| | Coef | Std err | t | P > [t} | 0.025 | 0.975 |
| Const | $-3640.5467$ | 845.613 | $-4.305$ | 0.000 | $-5298.422$ | $-1982.672$ |
| X-1 | 0.0002 | $4.22 \times 10^{-5}$ | 4.263 | 0.000 | $9.72 \times 10^{-5}$ | 0.000 |
| X-2 | $-0.5834$ | 0.011 | $-51.475$ | 0.000 | $-0.606$ | $-0.561$ |
| X-3 | 0.9383 | 0.011 | 88.633 | 0.000 | 0.918 | 0.959 |
| X-4 | 0.6442 | 0.010 | 64.914 | 0.000 | 0.625 | 0.664 |
| X-5 | $-1.319 \times 10^{-9}$ | $1.86 \times 10^{-9}$ | $-0.709$ | 0.478 | $-4.97 \times 10^{-9}$ | $2.33 \times 10^{-9}$ |
| X-6 | $3.868 \times 10^{-11}$ | $1.59 \times 10^{-11}$ | 2.432 | 0.015 | $7.49 \times 10^{-12}$ | $6.99 \times 10^{-6}$ |
| X-7 | $-2.888 \times 10^{-6}$ | $6.12 \times 10^{-7}$ | -4.724 | 0.000 | $-4.09 \times 10^{-6}$ | $-1.69 \times 10^{-6}$ |
| X-8 | 0.5162 | 0.045 | 11.540 | 0.000 | 0.429 | 0.604 |
| | Omnibus: | 1759.602 | Durbin–Watson: | 2204 | | |
| | Prob (Omnibus): | 0.000 | Jarque–Bera (JB) | 193,715.464 | | |
| | Skew: | 1.124 | Prob(JB) | 0.00 | | |
| | Kurtosis: | 37.078 | Cond. No. | $4.31 \times 10^{14}$ | | |

The variable x contains all 3986 rows and 9 columns of the data (attributes, e.g., 0, 1, 2, 3, 4, 5, 6, 7, 8). In Table 6, the constant x5 has the highest *p*-value of 0.478 compared to other constants, and is also higher than the defined significance level of 0.01. Thus, x5 is eliminated. In Step 2, the backward elimination method is repeated with the remaining constants, and the results are shown in Table 7.

**Table 7.** Backward elimination (Step 2).

| DEP.Variable | | Close | | R-Squared: | | 1.000 |
|---|---|---|---|---|---|---|
| Model: | | OLS | | ADJ. R-Squared: | | 1.000 |
| Method | | Least Squares | | F-statistic: | | $5.470 \times 10^6$ |
| Date: | | Sat, 31 July 2021 | | Prob (F-Statistic): | | 0.00 |
| Time: | | 12:20:45 | | Log-Likelihood: | | $-19,235$ |
| No. observations: | | 3986 | | AIC: | | $3.849 \times 10^4$ |
| Df Residuals: | | 3978 | | BIC: | | $3.854 \times 10^4$ |
| Df Model | | 7 | | | | |
| Covariance Type: | | Non-robust | | | | |
| | Coef | Std err | t | P > [t} | 0.025 | 0.975 |
| Const | $-3597.0316$ | 843.330 | $-4.265$ | 0.000 | $-5250.431$ | $-1982.632$ |
| X-1 | 0.0002 | $4.21 \times 10^{-5}$ | 4.224 | 0.000 | $9.53 \times 10^{-5}$ | 0.000 |
| X-2 | $-0.5839$ | 0.011 | $-51.624$ | 0.000 | $-0.606$ | $-0.562$ |
| X-3 | 0.9389 | 0.011 | 88.925 | 0.000 | 0.918 | 0.960 |
| X-4 | 0.6442 | 0.010 | 64.919 | 0.000 | 0.625 | 0.664 |
| X-5 | $3.528 \times 10^{-11}$ | $1.52 \times 10^{-11}$ | 2.324 | 0.020 | $5.54 \times 10^{-12}$ | $6.5 \times 10^{-11}$ |
| X-6 | $-3.007 \times 10^{-6}$ | $5.89 \times 10^{-7}$ | $-5.107$ | 0.000 | $-4.16 \times 10^{-6}$ | $-1.85 \times 10^{-6}$ |
| X-7 | 0.5113 | 0.044 | 11.572 | 0.000 | 0.425 | 0.598 |
| | Omnibus: | 1760.540 | Durbin–Watson: | 2205 | | |
| | Prob (Omnibus): | 0.000 | Jarque–Bera (JB) | 193,064.932 | | |
| | Skew: | 1.126 | Prob(JB) | 0.00 | | |
| | Kurtosis: | 37.020 | Cond. No. | $4.30 \times 10^{14}$ | | |

In Table 7, x contains all the rows, and the columns are [0,1,2,3,4,6,7,8]. After confirming the values in the backward elimination, x5 (i.e., 6th column) again showed the highest *p*-value of 0.020, compared to the other constants, and it is above the significance level of 0.01. Thus, x5 is eliminated. Again, the process is repeated with the remaining variables.

In Table 8, x contains all the rows, and the columns are [0,1,2,3,4,7,8]. After confirming the values in the backward elimination, all the constant's *p*-values are less than the significance level. Thus, we need to stop the backward elimination process. The output of the more correlated features identified using the backward elimination method is shown in Table 9.

**Table 8.** Backward elimination (Step 3).

| DEP.Variable | | Close | | R-Squared: | | 1.000 |
|---|---|---|---|---|---|---|
| Model: | | OLS | | ADJ. R-Squared: | | 1.000 |
| Method | | Least Squares | | F-statistic: | | $6.37 \times 10^6$ |
| Date: | | Sat, 31 July 2021 | | Prob (F-Statistic): | | 0.00 |
| Time: | | 12:20:45 | | Log-Likelihood: | | $-19,238$ |
| No. observations: | | 3986 | | AIC: | | $3.849 \times 10^4$ |
| Df Residuals: | | 3979 | | BIC: | | $3.854 \times 10^4$ |
| Df Model | | 6 | | | | |
| Covariance Type: | | nonrobust | | | | |
| | Coef | Std err | t | P > [t} | 0.025 | 0.975 |
| Const | $-2510.4851$ | 702.538 | $-3.573$ | 0.000 | $-3887.853$ | $-1133.117$ |
| X-1 | 0.0001 | $3.5 \times 10^{-5}$ | 3.524 | 0.000 | $5.48 \times 10^{-5}$ | 0.000 |
| X-2 | $-0.5835$ | 0.011 | $-51.565$ | 0.000 | $-0.606$ | $-0.561$ |
| X-3 | 0.9381 | 0.011 | 88.847 | 0.000 | 0.917 | 0.959 |
| X-4 | 0.6453 | 0.010 | 65.081 | 0.000 | 0.626 | 0.665 |
| X-5 | $-1.773 \times 10^{-6}$ | $2.56 \times 10^{-7}$ | $-6.920$ | 0.020 | $-2.28 \times 10^{-6}$ | $-1.27 \times 10^{-6}$ |
| X-6 | 0.5284 | 0.044 | 12.121 | 0.000 | 0.443 | $-0.614$ |

**Table 8.** *Cont.*

| | | | |
|---|---|---|---|
| Omnibus: | 1771.618 | Durbin–Watson: | 2.199 |
| Prob (Omnibus): | 0.000 | Jarque–Bera (JB) | 199,512.478 |
| Skew: | 1.132 | Prob(JB) | 0.00 |
| Kurtosis: | 37.585 | Cond. No. | $3.16 \times 10^{10}$ |

**Table 9.** More correlated features were identified using backward elimination.

| Constant | Attributes/Column Name |
|---|---|
| X1 | DATE |
| X2 | OPEN |
| X3 | HIGH |
| X4 | LOW |
| X7 | TRADE |
| X8 | RSI |

In the current study, the selected variables are then fed to the LSTM model, and its stock prediction accuracy is calculated. Again, to validate and compare the proposed model's effectiveness, its accuracy is compared with the output of the LSTM model, without using the backward elimination method, i.e., all the variables were fed into the LSTM model as input. While designing the LSTM model, two hidden layers were utilized with the "ReLU" activation function. The benefit of utilizing this ReLU function is that it does not trigger all the neurons at once. Hence, it takes less time to process. In the first hidden layer, 64 nodes are used, and in the second hidden layer, 32 nodes are used. Therefore, the total trainable parameters in this model is 330,369, as shown in Figure 4.

```
Model: "sequential"

_________________________________________________________________
Layer (type)                 Output Shape              Param #
=================================================================
lstm (LSTM)                  (None, 14, 64)            17920

_________________________________________________________________
lstm_1 (LSTM)                (None, 32)                12416

_________________________________________________________________
dropout (Dropout)            (None, 32)                0

_________________________________________________________________
dense (Dense)                (None, 1)                 33
=================================================================
Total params: 30,369
Trainable params: 30,369
Non-trainable params: 0
_________________________________________________________________
```

**Figure 4.** LSTM trained model.

To fit the model, we have considered 1030 and 50 epochs, with a batch size of 16. We observed the model performances by varying the epochs, and the performance measures (MSE, RMSE, and MAPE) have been calculated in each case. The classification results are shown in Table 10. While comparing the performance of the LSTM model before and after employing the backward elimination method, it was observed that the backward elimination method improved the classification performance significantly. Moreover, the accuracy in our proposed method has also been compared with the accuracy of some methods used in the previously reported literature (Table 11), in which several classification

models have been used. Ariyo et al. (2014) used the ARIMA model and achieved an accuracy of 90%, a precision of 91%, and a recall of 92%. Khaidem et al. (2016) utilized the random forest algorithm and achieved an accuracy of 83%, a precision of 82%, and a recall of 81%. Asghar et al. (2019) built a multiple regression model and achieved an accuracy of 94%, a precision of 95%, and a recall of 93%. Finally, Shen and Shafiq 2020 utilized FE+ RFE+PCA+LSTM and achieved an accuracy of 93%, a precision of 96%, and a recall of 96%. To our surprise, we noted the optimum performance in the proposed method, with high accuracy, precision, and recall scores, i.e., 95%, 97%, and 96%, respectively.

**Table 10.** Performance of backward elimination LSTM (BE-LSTM) compared with that of LSTM.

| Models | Epochs | Training Error | Validation Error | MSE | RMSE | MAPE |
|---|---|---|---|---|---|---|
| LSTM | 10 | 0.0192 | 0.0229 | 2,493,098 | 1578.95 | 10.66 |
| | 30 | 0.01 | 0.0095 | 1,826,370 | 1351.43 | 9.05 |
| | 50 | 0.0167 | 0.0178 | 1,232,070 | 1109.98 | 6.62 |
| Backward Elimination with LSTM | 10 | 0.0180 | 0.0171 | 465,470 | 682.25 | 5.33 |
| | 30 | 0.0154 | 0.0165 | 393,554 | 627 | 4.55 |
| | 50 | 0.0148 | 0.0157 | 383,597 | 619.35 | 3.54 |

**Table 11.** Comparison of the proposed solution with those in related works.

| Related Works | Models | Accuracy | Precision | Recall |
|---|---|---|---|---|
| (Ariyo et al. 2014) | ARIMA | 0.90 | 0.91 | 0.92 |
| (Khaidem et al. 2016) | Random Forest | 0.83 | 0.82 | 0.81 |
| (Asghar et al. 2019) | Multiple Regression | 0.94 | 0.95 | 0.93 |
| (Shen and Shafiq 2020) | Feature Expansion + Feature Selection + Principal Component Analysis + Long Short-Term Memory (FE+ RFE+PCA+LSTM) | 0.93 | 0.96 | 0.96 |
| Proposed method | LSTM | 0.84 | 0.83 | 0.84 |
| | **Backward Elimination with LSTM** | **0.95** | **0.97** | **0.96** |

Figure 5 shows the closing price of the NIFTY 50 index, and Figure 6 shows that in the 50 epochs model, the training loss is 1.5%, and the validation loss is 2.5%. Hence, the Be-LSTM model's performance is suitable for forecasting the future price of the NIFTY 50 stock. Figure 6 indicates that the data were taken as a look back period, where n = 3986 produced the outcome with a training loss of 1.5% and a validation loss of 2.5%. The LSTM and BE-LSTM testing mechanism is a reliable model for forecasting securities prices. The historical data from the NIFTY 50 index provided realistic output after testing on both the models, indicating a more reliable prediction using the BE-LSTM, with a standard deviation occurrence of 5% in the given sample size, as real independent data.

The input data, including high, open, low, and close, with relative strength index (RSI) and trades, trained the model to predict the future price over the next 30 days for the NIFTY 50 index. The model's accuracy suggests that it has achieved a good outcome for predicting the closing price of the NIFTY 50 index. The BE-LSTM is more accurate than the LSTM for price prediction.

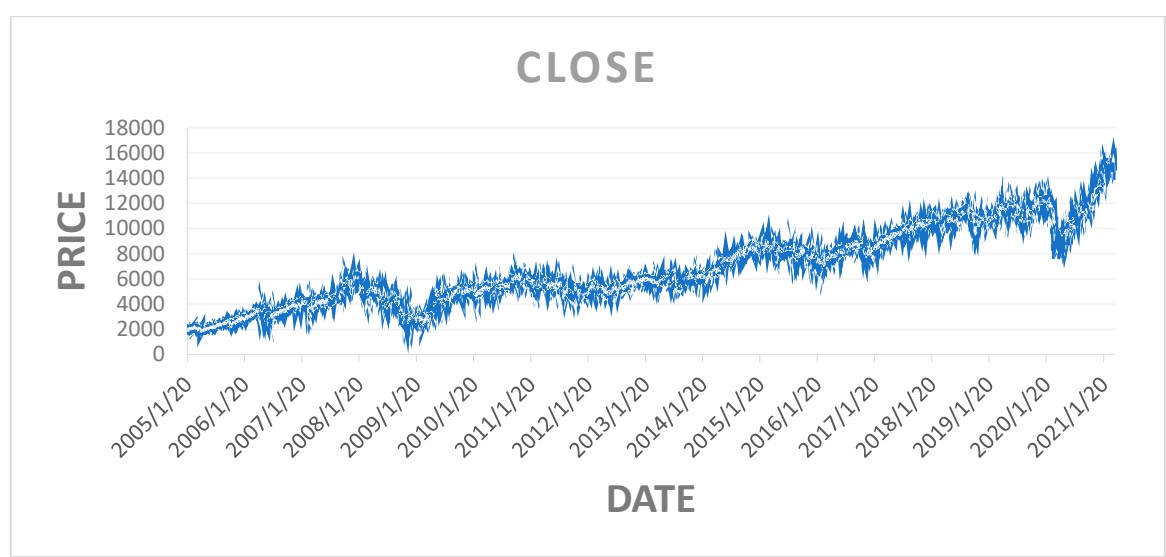

**Figure 5.** Closing price of the NIFTY 50 index.

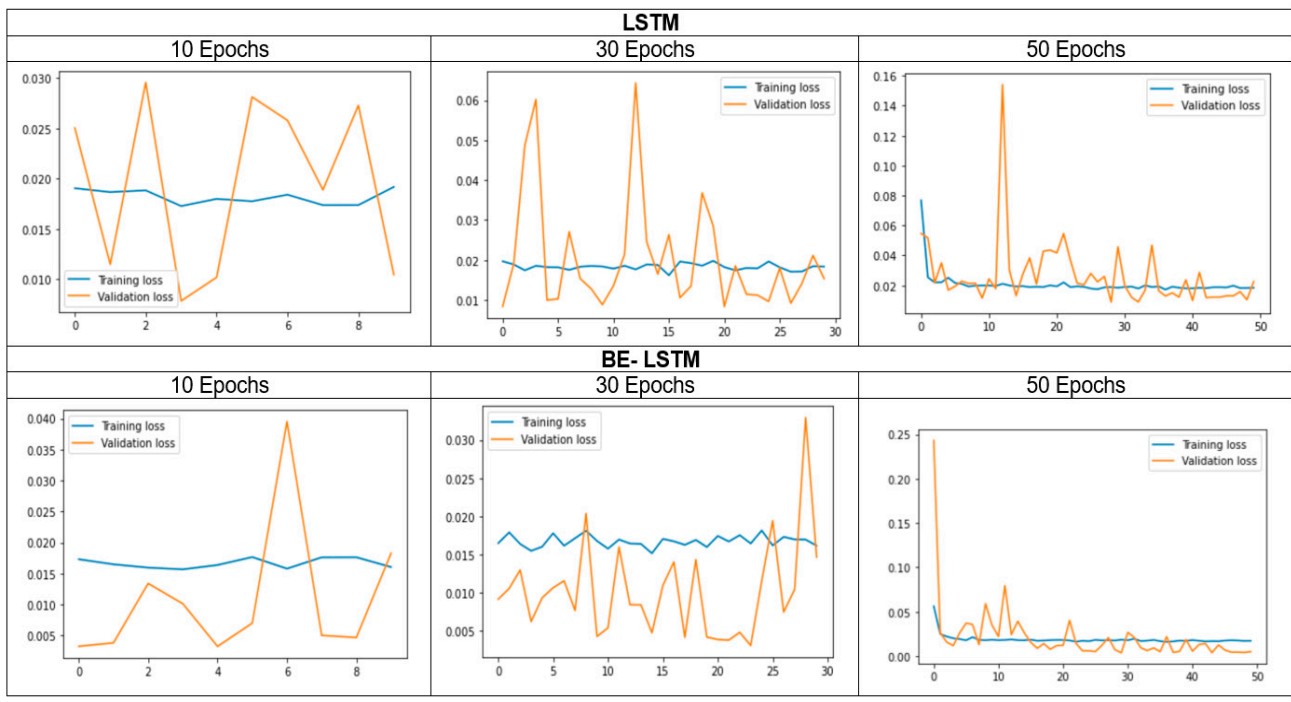

**Figure 6.** Training and validation loss of both LSTM and BE-LSTM.

The standard deviation of the outcome, compared with the input data and the validation, indicates the number of errors in the model. We considered 3986 data points in the model analysis to train the LSTM and the backward elimination with LSTM. The significant output reflects the average epochs of around 5%, which are considerable remarks for building confidence in the tested model. From Figure 6, we observed that the proposed BE-LSTM performs well compared to the LSTM model. This is because BE-LSTM helps in eliminating the irrelevant features or input nodes, reducing complexity and enhancing efficiency, yielding faster training times and reduced memory requirements, features which are lacking in LSTM. Secondly, as irrelevant nodes are eliminated, the remaining features become more important for capturing the temporal dependencies of the sequence. This helps to provide insights into which features contribute significantly to the model's predictions. By understanding the influential features, we can make more informed decisions, identify critical factors, and improve the overall interpretability of the model.

Finally, Figure 7 compares three closing prices from 5 March 2021 to 31 March 2021. It clearly depicts that BE-LSTM performs well compared to the conventional LSTM model.

**Figure 7.** The closing price for the next 30 days (differences between the original close value, the LSTM, and the BE-LSTM).

## 5. Conclusions and Future Scope

The emerging technology in the financial field, along with its combination with artificial intelligence, is an evolving area of research. This paper proposes a more suitable AI-based method rather than the traditional approach (fundamental analysis, technical analysis, and data analysis) for predicting the NIFTY 50 index price for the next 30 days using the BE-LSTM model.

The dimensional work for determining the NIFTY 50 index price showcases the comparison of LSTM and BE-LSTM for an equilateral dataset. In this work, the BE-LSTM, whose results are much closer to the original close price, is gaining favor in the area of stock price prediction. At the same time, the LSTM showed a deviation in predicting the output when compared to the actual price. The results suggest that the BE-LSTM model showed improved accuracy compared to the LSTM method. In the future, the backward elimination method can be employed with other deep learning methods, such as GAN with varied hyperparameters, for investigating alternative algorithm improvements.

The financial industry is now inclining towards the adoption of technology in various areas, including portfolio management, wealth management, equity analysis, and derivative research. The brokerage houses, as well as fund management and portfolio management services, have struggled to analyze asset prices. This study will help those involved in the finance industry, along with policy makers, to use emerging technology like artificial intelligence in finance. It will also aid the policy makers in analyzing the market sentiment and trends using appropriate algorithmic trading, employing predictive models to create investor awareness and enhance the number of market participants. It is crucial for regulators and policy makers to understand the volatility of the stock market in order to steer the economy toward development, to ensure the smooth operation of the stock exchange, and to encourage more investors—particularly retail investors—to engage in the market. As a result, stronger investor protection measures, as well as more investor education initiatives, will be adopted.

In addition, investors want to generate a significant return on a less risky investment. Therefore, before making an investment decision, Indian investors are required to carefully

study and analyze the stock market volatility using publicly accessible information, as well as many other impacts, as this analysis is essential for determining the effectiveness and volatility of stock markets. This study will help investors manage risk by identifying potential market downturns through artificial intelligence, enabling the adjustment of portfolios and the minimization of loss.

**Author Contributions:** Conceptualization: all authors; methodology: all authors; software: all authors; validation: all authors; formal analysis: all authors; investigation: all authors; resources: all authors; data curation: all authors; writing—original draft preparation: all authors; writing—review and editing: all authors; visualization: all authors; supervision: S.H.J.; project administration: S.H.J. All authors have read and agreed to the published version of the manuscript.

**Funding:** This research received no external funding.

**Institutional Review Board Statement:** Not applicable.

**Informed Consent Statement:** Not applicable.

**Data Availability Statement:** Data are available from El-Chaarani upon request.

**Conflicts of Interest:** The authors declare no conflict of interest.

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
