# Peer review of "Forecasting of NIFTY 50 Index Price by Using Backward Elimination with an LSTM Model"

_jrfm, doi:10.3390/jrfm16100423_

Round 1
Reviewer 1 Report
Thank you for the opportunity to review “Forecasting of NIFTY50 Price by Using Backward Elimination 2 With LSTM Model”.
After reading the title, I was expecting some new evidence with significant implications for investors and policy makers. However, after reading this article, I did not find any significant contribution for the readers.
The comparison between conventional and AI tools is not discussed in detail. It’s just mentioned in a single line. The authors should discuss various conventional models and add some criticism in case of stock market prediction.
Figure 1 shows the Convolutional BE-LSTM process. This Figure is not even cited anywhere in the paper. What is the purpose of this Figure. If this is the framework, it is very much different from the real model.
Figure 3 shows the very basic steps and ignored many relevant steps to perform LSTM model. It should be detailed and discussed in the methodology.
What is the rationale behind selection of variables Open, High, Low, Close, Volume, Value, Trades, RSI and RSI Average. The indication RSI and RSI average include the information of other variables already. The selection must be logical with proper references.
Mention the names of variables in elimination method. X-1, X-2 not clear.
Firstly, Figure 5 should be a table. What does figure 5 shows? It is just mentioned that these are “MORE Correlated” and no more discussion.
What is the data range?
The captions of the figures are not clear.
What are the implications of this study?
In current form, this is a good exercise but not suitable for journal Publication. Overall, this study needs serious changes in writing, exposition, methodology and conclusions.
NA
Author Response
Reviewer 1
Thank you for the opportunity to review “Forecasting of NIFTY50 Price by Using Backward Elimination 2 With LSTM Model”.
Comment: After reading the title, I was expecting some new evidence with significant implications for investors and policy makers. However, after reading this article, I did not find any significant contribution for the readers.
Sisodia, P. S., Gupta, A., Kumar, Y., & Ameta, G. K. (2022, February). Stock market analysis and prediction for NIFTY50 using LSTM Deep Learning Approach. In 2022 2nd International Conference on Innovative Practices in Technology and Management (ICIPTM) (Vol. 2, pp. 156-161). IEEE.
Mahajan, V., Thakan, S., & Malik, A. (2022). Modeling and forecasting the volatility of NIFTY 50 using GARCH and RNN models. Economies, 10(5), 102.
Zaheer, S., Anjum, N., Hussain, S., Algarni, A. D., Iqbal, J., Bourouis, S., & Ullah, S. S. (2023). A Multi Parameter Forecasting for Stock Time Series Data Using LSTM and Deep Learning Model. Mathematics, 11(3), 590.
Sharma, D., Sarangi, P. K., & Sahoo, A. K. (2023, May). Analyzing the Effectiveness of Machine Learning Models in Nifty50 Next Day Prediction: A Comparative Analysis. In 2023 3rd International Conference on Advance Computing and Innovative Technologies in Engineering (ICACITE) (pp. 245-250). IEEE.
Zeng, X., Cai, J., Liang, C., & Yuan, C. (2023). Prediction of stock price movement using an improved NSGA-II-RF algorithm with a three-stage feature engineering process. Plos one, 18(6), e0287754.
Comment: The comparison between conventional and AI tools is not discussed in detail. It’s just mentioned in a single line. The authors should discuss various conventional models and add some criticism in case of stock market prediction.
Ans: Due to its inherent complexity and uncertainty, stock market forecasting frequently draws criticism. Some claim that focusing entirely on technical analysis and historical data might leave out important outside influences. Predictions are unreliable because stock prices, according to the efficient market hypothesis, already reflect all available information. Additionally, it's possible for human nature and market emotion to be unexpected and unreasonable. The possibility for bias and overfitting in prediction algorithms is highlighted by critics. Although improvements in machine learning have increased accuracy, doubts still exist over the capacity to anticipate market moves consistently, highlighting the speculative nature of stock forecasting.
Based on the lictrature review it is clearly understood that all those papers have used various methodology to forcast the stock price, but no one has discussed the approach on identifying the more correlated features before passing into the LSTM model for stock price prediction. To identify the best corelated features, we have built Backward Elimination with LSTM (BE-LSTM) for predicting the stock price, and also, we made a comparison with the general LSTM without the backward Elimination approach to check which model is the best.
Figure 1 shows the Convolutional BE-LSTM process. This Figure is not even cited anywhere in the paper. What is the purpose of this Figure. If this is the framework, it is very much different from the real model.
Ans: As the framework is mention in the below section in 3.4. We have removed the convolutional BE- LSTM process figure.
Comment: Figure 3 shows the very basic steps and ignored many relevant steps to perform LSTM model. It should be detailed and discussed in the methodology.
Ans: The figure is updated in the paper
Comment: What is the rationale behind selection of variables Open, High, Low, Close, Volume, Value, Trades, RSI and RSI Average. The indication RSI and RSI average include the information of other variables already. The selection must be logical with proper references.
Ans: The variables are selected from the technical analysis daily trading data. The most relevant data points in technical analysis are HOLC (High, Open, Low & Close) followed by Volume, trade and others. All the indicators like RSI, MACD, stochastic value is derived from the HOLC data based on their calculation and parameters. The RSI is most widely used and most appreciated and acceptable indicators across the globe. Hence, we used RSI data points along with HOLC. RSI Average is being ignored before running the data in the model.
RSI Calculation:
RSI= 100-100/1+RS
RS= Average Gain/ Average Loss
Here, look back period is 14 days for RSI calculation
Comment: Mention the names of variables in elimination method. X-1, X-2 not clear.
Ans: At the time of passing data in the BE process the constant is assign as 1, X-1 represents date, X-2 represents Open, X-3 represent High and so on… representing the index value of the columns in the dataset below:
|
Const |
Attributes/ Column name |
|
X1 |
DATE |
|
X2 |
OPEN |
|
X3 |
HIGH |
|
X4 |
LOW |
|
X5 |
VOL |
|
X6 |
VALUE |
|
X7 |
TRADE |
|
X8 |
RSI |
In the case of backward elimination, we are now attempting to remove less important variables from the model. It usually entails fitting the model repeatedly, determining the importance of each variable, and eliminating the least relevant ones.
In essence, we enable the model to include an intercept term const that reflects the projected value of y when all independent variables are set to zero by including a constant term (a column of 1s) in the dataset. When the independent variables have no influence, the baseline level of y is captured by this intercept term.
Because when a variable is removed from the model, we are effectively determining its relevance by observing how the overall model performance (often evaluated by a metric like p-value, AIC, or R-squared) changes, adding this constant term is very important during backward elimination. Without the constant term, eliminating a variable can lead to a model that assumes the dependent variable starts at zero in the absence of all other factors, which may not be applicable in many real-world cases.
Comment: Firstly, Figure 5 should be a table. What does figure 5 shows? It is just mentioned that these are “MORE Correlated” and no more discussion. The captions of the figures are not clear.
Ans:
|
Const |
Attributes/ Column name |
|
X1 |
DATE |
|
X2 |
OPEN |
|
X3 |
HIGH |
|
X4 |
LOW |
|
X7 |
TRADE |
|
X8 |
RSI |
Comment: What is the data range?
The data range selected for the study is taken from the Bloomberg data base start from 11 Feb 2005 and last data is till 5 Mar 2021. The data consideration is of approximately 15 years which consist of Bull and Bear phase of Indian Equity market for better analysis and prediction.
Comment: What are the implications of this study?
The Finance industry is now inclining towards adoption of technology in various area like Portfolio management, Wealth Management, Equity analysis, Derivative research, Fundamental and Technical Analysis and trading and many more. The broking houses, fund management and Portfolio management services struggled for analysing the asset price. This study will help finance industry to use emerging technology like Artificial Intelligence in finance. The proposed study is more suitable deep learning based method rather than the traditional approach (fundamental analysis, technical analysis or Data analysis) for predicting asset price.
The study compares the two deep learning model like LSTM and BE-LSTM and signing mark in favour of BE-LSTM, which predict the price more closer to real price.
Reviewer 2 Report
It was demonstrated that forecasting the NIFTY50 Price using LSTM and BE-LTSM was fruitful. This is consistent with the expected power and utility of artificial intelligence-based techniques for stock market prediction.
I recommend giving mathematical formulas for calculating the accuracy, precision and recall characteristics.
Quantification of prediction success can be improved. Adding characteristics of Win ratio, profit loss ratio, profit factor, and profitability would give a better insight into the success of the prediction. Cf. The Profitability Rule. URL: http://www.priceactionlab.com/Literature/profitability.pdf
It needs to be made clear from the text which variables and from which times are used as explanatory variables in the model. Unfortunately, references to using HOLC and RSI without the specified details are insufficient.
I need help understanding the application of the Z-test in the hypothesis of equality of probabilities H0: p=s/n = p0=s0/n. /line 215/
When describing the LSTM, I am missing information about the vector length assigned to the node. /line 223/
The function/matrix P_e, P_j, P_m is not explained. /lines 259,265, 266/
What is the difference between w and W? /line 259,265,266 x 268/
Commas and a dot are missing after formulas (6), (7), and (8).
Tab. 5, what is an omnibus?
Tab. 5, 6, 7: Instead of const., X1, X2, ..., X8, it is more appropriate to write the estimate \hat{beta}_0, \hat{beta}_1, ..., \hat{beta}_7.
L-Jung box test /Ljung–Box test; cite a reference
augmented Dickey–Fuller test; reference
Author Response
Reviewer 2
It was demonstrated that forecasting the NIFTY50 Price using LSTM and BE-LTSM was fruitful. This is consistent with the expected power and utility of artificial intelligence-based techniques for stock market prediction.
Comment: I recommend giving mathematical formulas for calculating the accuracy, precision and recall characteristics.
Ans: Accuracy
Accuracy serves as a metric that offers a broad overview of a model's performance across all classes. It proves particularly valuable when all classes hold equal significance. This metric is computed by determining the proportion of correct predictions in relation to the total number of predictions made.
Calculating accuracy with Scikit-learn based on the previously computed confusion matrix is as follows: We store the result in the variable accuracy by dividing the sum of True Positives and True Negatives by the sum of all values within the matrix.
Precision
Precision is computed by taking the proportion of correctly classified Positive samples relative to the total number of samples classified as Positive, whether correctly or incorrectly. Precision serves as a metric for gauging the model's accuracy when it comes to classifying a sample as Positive.
When the model generates numerous incorrect Positive classifications or only a few correct Positive classifications, this elevates the denominator and results in a lower precision score. Conversely, precision is higher under the following conditions:
The model produces a substantial number of correct Positive classifications, thus maximizing True Positives.
The model minimizes the number of incorrect Positive classifications, thereby reducing False Positives.
Recall
Recall is determined by the ratio of correctly classified Positive samples to the total number of Positive samples. It quantifies the model's capacity to identify Positive samples. A higher recall score signifies a greater ability to detect Positive samples.
Recall exclusively focuses on the classification of positive samples and is independent of the classification of negative samples, as observed in precision. If the model categorizes all positive samples as Positive, even if it incorrectly labels all negative samples as Positive, the recall will still register at 100%.
Comment: Quantification of prediction success can be improved. Adding characteristics of Win ratio, profit loss ratio, profit factor, and profitability would give a better insight into the success of the prediction. Cf. The Profitability Rule. URL: http://www.priceactionlab.com/Literature/profitability.pdf
Ans: This Paper is developed as a comparative analysis of two Deep learning model LSTM and BE- LSTM. To analyse the suitable model for predicting price closer to real time price. However, the same work is being done for global data in my another research work along with P/L calculation, profit factor in another research work which is extensions of the paper and under progress.
Comment: It needs to be made clear from the text which variables and from which times are used as explanatory variables in the model. Unfortunately, references to using HOLC and RSI without the specified details are insufficient.
|
Ans: Const |
Attributes/ Column name |
|
X1 |
DATE |
|
X2 |
OPEN |
|
X3 |
HIGH |
|
X4 |
LOW |
|
X5 |
VOL |
|
X6 |
VALUE |
|
X7 |
TRADE |
|
X8 |
RSI |
Comment: I need help understanding the application of the Z-test in the hypothesis of equality of probabilities H0: p=s/n = p0=s0/n. /line 215/
Ans: Is implemented in paper.
Comment: When describing the LSTM, I am missing information about the vector length assigned to the node. /line 223/
Ans: The vector length is 64. Is included in paper.
Comment: The function/matrix P_e, P_j, P_m is not explained. /lines 259,265, 266/
Ans: Pe represents weight matrix associated with the forget gate
Pj represents weight matrix associated with input gate
Pm represents weight matrix associated with output gate
Is included in paper.
Comment: What is the difference between w and W? /line 259,265,266 x 268/
Ans: Is typo error all are capital.
Comment: Commas and a dot are missing after formulas (6), (7), and (8).
Ans: Is Included
Comment: Tab. 5, what is an omnibus?
Ans: The word "omnibus" in backward elimination model selection often denotes a statistical test or metric that evaluates the overall fit of a regression model prior to deleting predictor variables. It aids in determining if the model is statistically significant in explaining the fluctuation of the dependent variable as a whole.
Comment: Tab. 5, 6, 7: Instead of const., X1, X2, ..., X8, it is more appropriate to write the estimate \hat{beta}_0, \hat{beta}_1, ..., \hat{beta}_7.
Ans: instead X1, Beta1
Comment: L-Jung box test /Ljung–Box test; cite a reference
Ans: added
Idrees, S. M., Alam, M. A., & Agarwal, P. (2019). A prediction approach for stock market volatility based on time series data. IEEE Access, 7, 17287-17298.
augmented Dickey–Fuller test; reference
Vaisla, K. S., & Bhatt, A. K. (2010). An analysis of the performance of artificial neural network technique for stock market forecasting. International Journal on Computer Science and Engineering, 2(6), 2104-2109.

Round 2
Reviewer 1 Report
This version is much better than the previous with significant improvements.
The exposition of the paper is still not standard. The use of different fonts, table formats, and images makes it unprofessional. It should be improved.
I am not convinced by the implications of the study. The implications should be linked with the forecasting of asset prices. How can investors make investment/risk management decisions in light of these findings? Likewise, How these findings are helpful for policy-making.
Author Response
This version is much better than the previous with significant improvements. The exposition of the paper is still not standard. The use of different fonts, table formats, and images makes it unprofessional. It should be improved.
Ans: The fonts, table format and images is improved as per suggestion.
Comment 2
Ans: I am not convinced by the implications of the study. The implications should be linked with the forecasting of asset prices. How can investors make investment/risk management decisions in light of these findings? Likewise, How these findings are helpful for policy-making.
Ans: The finance industry is now inclining towards adoption of technology in various area like portfolio management, wealth management, equity analysis, derivative research. The broking houses, fund management and portfolio management services struggled for analysing the asset price. This study will help finance industry and the policy makers to use emerging technology like Artificial Intelligence in finance. It will also help the policy makers in analyzing the market sentiment and trends with appropriate algorithmic trading using predictive models to create investors awareness and enhance the frequency of market participants. It is crucial for regulators and policymakers to understand the volatility of the stock market in order to steer the economy toward development and ensure the smooth operation of the stock exchange and to encourage more investors to engage in the market, particularly retail investors, stronger investor protection measures and more investor education initiatives will be adopted.
In addition, Investors want to generate a significant return on a less risky investment. Therefore, before making an investment decision, Indian investors are required to carefully study and analyse the stock market volatility using publicly accessible information and many other aspects because it is essential for determining the effectiveness and volatility of stock markets. This study will help the investors in managing risk by identifying potential market downturns through artificial intelligence which will help to adjust portfolios and minimize loss.